# Influence of Experiential Consumption and Social Environment of Local Tourists on the Intention to Revisit Tunisian Guesthouses: Mediating Role of Involvement in the Experience

**Nesrine Khazami [1,\*] and Zoltan Lakner [2]**

[1] Doctoral School of Management and Business Administration, Hungarian University of Agriculture and Life Science (MATE), 2100 Godollo, Hungary

[2] Department of Food Economics, Faculty of Food Science, Hungarian University of Agriculture and Life Science (MATE), 1118 Budapest, Hungary; Lakner.Zoltan.Karoly@uni-mate.hu

\* Correspondence: Khazami.Nesrine@phd.uni-mate.hu

**Abstract:** This study aims to examine the relationships between the experiential consumption, the social environment, and intention to revisit. The mediating role of involvement in the experience between experiential consumption and the intention to revisit and between the social environment and the intention to revisit has been expressed and tested. The data for this research was collected from local tourists staying in guesthouses scattered all over Tunisia. The authors applied structural partial least squares equation modeling to analyze 259 questionnaires completed by participants and to test the hypotheses. The authors found a positive and direct effects of the social environment on involvement in the experience while experiential consumption did not. In addition, the results indicated positive and significant indirect effects for the social environment on the intention to revisit through involvement in the experience. The results do not support a mediating role of involvement in experience on the relationship of experiential consumption and intention to revisit. In addition, the results showed a strong and positive effect of involvement in the experience on intention to revisit. This research makes a distinctive theoretical contribution to the literature of perceived experiential value by analyzing the relationships between experiential consumption and the social environment on experience involvement and intention to revisit a guesthouse. In addition, this study explores several practical implications of these results.

**Keywords:** experiential consumption; social environment; involvement in the experience; intention to revisit

## 1. Introduction

Tourist experience refers to "a constructivist conception of tourist reality in which individuals are fully involved in what they experience during their travels" [1]. These are tourist experiences based on breaking out of the everyday and interacting with the other and the unknown. Tourist experiences can be considered globally (a trip, a stay) or selectively (a practice, a product, or a service). They include what the tourist experiences according to his initial motivations and perceptions, in the context of activities carried out before, during, and after a stay.

This approach to tourist phenomena through the concept of tourist experiences has recently attracted the attention of researchers [1,2]. The creation and management of quality experiences for tourists have become an important strategic focus of the development of tourism businesses and providers of their products and services. The demand of tourists to have not only a memorable experience but also a quality experience is a real challenge for tourism stakeholders. Researchers increasingly point out that tourists are interested in products and/or services that allow them to create their own experiences linked to their feelings, identity, or social status [3,4]. However, on the supply side, the tourist

experience results from the combination of activities, services, or products involving private and public actors [5]. It is also important that tourism experiences are articulated to meet specific market expectations, through increasingly careful engineering that involves the detailed choreography of activities, products, and services, and the consideration of the characteristics of the place by different local actors [6]. It is an identity-building process based on specific characteristics that are either traditional or have emerged with the development of the place, or even co-created with the involvement of different actors, including tourists.

Local uniqueness can be used as a resource to create unique and differentiated experiences for consumers, thus, creating a competitive identity for the place. This implies the uniqueness of the place [4], which adds value and contributes to tourists' memorable experiences [2]. Thus, the development of local typicality, which was part of a "natural" process of spatial differentiation, is now a process that must produce distinctive experiences for consumers and that creates a competitive identity for tourism managers.

In this sense, the various players have never ceased to compete with ingenuity to offer formulas, locations, new forms of accommodation that involve novelty, originality, quality contribution, adaptation to needs and offering undeniable benefits to stimulate and retain customers. However, with the emergence of tourist demand, Tunisian stakeholders have begun to promote a new accommodation formula: guesthouses. This trend phenomenon is developing in Tunisia to meet the needs of the growing number of tourists looking for alternatives other than regular tourism. The implication in this revolution of the tourism sector is a reaction against mass tourism and the search for a unique and perfectly personalized trip, outside the classic formulas [7], especially after the Covid 19 pandemic, which is one of the most prominent phenomena of the 21st century [8]. In this unprecedented situation, the brand image of the tourist destination is not immutable. It must change according to the socio-economic, political, historical, or health context of the destination. Thus, the image that tourists have of a country or a city at a certain moment may change a few years or even a few months later. This evolution, therefore, depends on the events that have taken place in the tourist destination. As a result, the actors of this sector try to develop new strategies to save it. The ability of local actors to control these events and their impact on local tourists seems to be a profitable solution that can improve local tourism, since Tunisians must be part of the tourist market to be developed.

This study aims to explore the tourists' experiential consumption and social environment that lead them to return to alternative accommodation; in particular, the case of guesthouses is deepened to answer the following problem: How do experiential consumption and social environment influence local tourists' intention to return to guesthouses?

Therefore, the aim of this study is to explore how Tunisian local travelers had a guesthouse experience. In addition, this study explores the relative importance of experience consumption and social environment in a guesthouse that provides deep involvement in the experience during the stay. This study assesses the relative importance of experiential lodging factors that translate into an increased likelihood of repeat visitors. To explore the objectives of this paper, this article is structured as follows: After this introduction, the literature on the variables that appear in our study must be presented. In the next part, the presumed relationships between these values will be investigated through empirical observation, and the results and implications will be discussed.

## 2. Literature Review

### 2.1. Tourism Experience

The design of a tourist experience is made up of two parts: a "hard" part includes the basic material and organizational elements of tourism, aiming to meet the general needs of tourists such as moving, eating, sleeping, etc. This part forms the "support experience" [9]. The "soft" part focuses on emotional aspects of value creation, motivating the tourist to participate in specific attractions. This "peak experience" part is more about

activities that are different from everyday life and contain elements of surprise and personal enrichment [4].

The "peak experience" part reflects the real motivations of tourists, leads them to get involved and, thus, most strongly affects their perceptions, feelings, or emotions, all of which makes the experience more meaningful to them. It is rather difficult to rigorously distinguish the support and peak experiences because they are partially interchangeable and strongly interwoven with each other. Hence the need to deliver an experiential level of value to make a stay a satisfying and enchanting experience.

### 2.2. Experiential Consumption Components

The tourist experience refers to all types of practices and experiences of tourists before, during, and after their trip. To better understand this concept, it is important to analyze the components of a tourism experience.

Chen and Rahman [10] tried to categorize the dimensions of the tourist experience. The components of the tourism experience are complex and vary widely in the literature [10,11]. We would like to summarize here the elements or components of the tourist experience depending on the approach with which it is approached. From a psychological point of view, Larsen [12] suggested that the concept of tourism experience includes expectations, events, and memories, opening up a perspective for studying the tourism experience over the long term. Lin and Kuo [11], studying the behavioral consequences of the experience lived in tourist townships, proposes a conceptual model of the tourist experience, based on three main components: Tourists once placed in a situation with (1) stimuli: sensory, affective, behavioral, intellectual, and relational are involved in a (2) flow experience and live (3) emotions. This study reveals a psychological process: tourist experience →perceived value →satisfaction →loyalty. Most recently, Pharino, Pearce, and Pryce [13] describe the facets of the tourist experience in paranormal sites by applying the Orchestra model, which includes five components: sensory, cognitive, affective, behavioral, and relational. This study shows that visitors to paranormal sites experienced both positive and negative emotions.

The psychological approach generally examines the significance, especially the long-term memorability of tourist experiences, as well as their impacts on subsequent behaviors of tourists. In short, studies on the tourist experience from a psychological angle are essentially part of behaviorism to give recommendations in terms of management or marketing in tourism. It is not surprising that the psychological aspects of tourist experiences are also mentioned in other approaches that we will see below.

The social science approach rather tightens the tourist experience on its "peak experience" dimension, while the marketing/management approach considers it more like a "consumer experience" [9]. The first approach focuses on the fact that the tourist seeks an experience that is different from their daily life, while the basic elements of a trip like accommodation, food, and transport are often ignored in studies [9]. The second views tourists as consumers in trading relationships and approaches the tourism experience as any type of product or service without considering the distinction between a peak experience and a support experience [14] According to this approach, the focus is on the quality of services [14]. Quan and Wang [9] combine the two approaches to propose a conceptual framework of the tourist experience combining "the peak experience" (escape from everyday life) and "support experiences" (eating, sleeping, moving around). Indeed, these peak and support experiences can be interchangeable [14]. In most cases, the quality of the support experiences can improve the peak experience, and they can also fill any perceived gaps in the peak experience, especially through a compensating effect regarding the perception of the total quality of the peak experience.

The marketing and management approach considers the tourism experience as a consumption experience [15]. Tourists as consumers are involved in various exchange of service relationships [14]. These transactions require the active participation of the tourist [16]. Of course, individual tourists react differently to the same events and stimuli.

To determine the relationships of the influencing factors of consumer experience, Mossberg [14] developed a conceptual model. The touristic attractions lead to extraordinary, memorable experiences, with contributions to the better positioning of touristic destination and enterprise in a competitive economic environment. However, seeing everything as experience leads to a kind of experiential "inflation", which ends up being a problem for providers, because when everyone is promoting the same or similar experiences, the ubiquity of the experience will be decreasing. The abuse of the term "experience" can also cause disappointment on the customer side. These findings show the need to study in-depth the elements that contribute to the formation of the extraordinary, meaningful, and memorable experience.

As we have seen, the tourism experience can be approached from different sides, but then it always remains a central problem of modern tourism research. "Quality tourism experience" is a term used repeatedly by all stakeholders of the tourism sector, including organizations involved in tourism research, planning, policy, management, marketing, and distribution [4]. The conceptualization of experiential quality encompasses the socio-psychological expectations and emotional responses of tourists to the tourism experience [17]. This concept covers a wider range of perspectives than the mere quality of service. The quality of a tourism experience is associated not only with the quality of the product and the relationship between quality and satisfaction, but also with the overall environment of the experience. The latter includes the geographical setting (natural or artificial), the identity, the reputation, and the sustainability of the place, the interactions between the host and the visitor, the type of experience, as well as the motivation of the tourist [4]. Experiential quality is rather hard to qualify, it can be in an indirect way, built on psychological outcome experienced by clients who have participated in tourism activities. The criteria for this measure derive from the propensity (and ability) of tourists to rate the quality of their experiences based on individual emotional responses rather than functional or utilitarian standards [17].

Several authors have tried to operationalize the quality of the tourist experience by proposing and validating different measurement scales [18–21]. For example, Otto and Ritchie [22] measured the quality of the service experience in tourism (airlines, hotels, tours, and attractions) across four dimensions: hedonism, peace of mind, involvement, and recognition. The uniqueness and/or the degree of differentiation of the experience have been listed as factors that contribute to the hedonic dimension of the experience. In their study, Kao and al. [20] and Jin and al. [19] identified four factors: immersion, surprise, participation, and pleasure, to assess the experiential quality in water parks and theme parks, which, depending on the case, represents an exciting or entertaining tourist attraction focused on a specific theme. Surprise refers to the uniqueness of an experience, usually created from exceptional stimuli in unpredicted circumstances when consuming tourism products or services. Moon and Han [21] recently examined the quality of tourism experiences of Chinese travelers compared to a well-known South Korean island-destination, as a mediator between destination attributes, satisfaction, and behavioral intentions. The evaluation of the quality of the visitor experience is based on four dimensions: hedonism, peace of mind, involvement, and escape. Local typicity such as the uniqueness of the architecture, the design of the infrastructure, the local culture, or the unique nature of the island were attributes of the destination. Results of the survey have shown that Chinese tourists felt less relaxed and more frustrated when they cannot control certain novel things. The unusual features from point of view of tourists at a certain level stimulate a sense of distance for travelers. This shows that the novelty contributes to memorability but not necessarily to the satisfaction with the tourist experience.

In short, the factors for analyzing a tourist experience can be clustered into three interactive groups:

- Environmental and situational factors: these are elements linked to the physical environment of the spatial and/or temporal context where the tourist experience

takes place. It can be a destination, a park, a hotel, etc. The timing and the general mood of that environment also play an important role in these factors.

- Internal factors of the experience itself: these are the emotions, sensations, or personal perception of the tourist such as immersive impression, surprise, escape, entertainment, etc., and the sense of being the main actor in the experience.
- Relational factors: these are the tourist's interactions with the environment, other tourists, the communities concerned such as residents, tourism stakeholders, and even activities, products or services, technologies, etc., in tourism during the experience.

The combination of these three categories contributes to the success of a tourist experience. The value of this tourist experience is multidimensional. The most important dimensions Aurier, Evrard and N'Goala [23] define as:

- Utility value: the lived experience retains a pragmatic, physical, practical, tangible actual value concerning the touristic products. This utility value can be further structured by different aspects: the functional [24], cognitive [25], financial [26] considerations, orientation towards the benefits of the object [27], the satisfaction of different levels of human needs according to Maslow pyramid [28].
- Social links: the lived experience lends itself for exchange, conversation, and social interaction. We find: behaviors [25], interpersonal links [26], social value [24], interpersonal orientation [27], social practices [29].
- Experiential stimulation: the consumption of the touristic product/service stimulates the consumer's senses in order to give him a real consumption experience where feelings and emotions are intertwined. We find: pleasure, imagination, affect [25], playful value [26,30], hedonistic value [24,29], aesthetic value [30], emotional value [24].
- Knowledge: the lived experience offers consumers the information and expertise necessary for the enrichment and for the control of its environment. We find: the epistemic value [24], the search for information, the subjective expertise [29].
- Self-expression: the lived experience develops the personal development of the consumer who enriches his identity creation by approaching the ideal self. We find: self-expression, self-realization [26], orientation towards oneself [27], esteem [30], sign value [29].
- Spiritual value: the lived experience allows consumers to question themselves and question themselves in the face of society, others and humanity in general. We find: spiritual value [26], spirituality, and ethical status [30].

This transversal synthesis will allow us to e mpirically test the experiential consumption of staying in a guesthouse. This study, therefore, seeks to shed light on a better understanding of the local tourist through the value they perceive from their accommodation experience in guesthouses.

### 2.3. Social Environment

Today, many customers search about the uniqueness. They are looking for a differentiated and more personalized service [31], beginning with hospitality, which creates an atmosphere in which visitors feel expected and desired. As Gouirand [32] writes, "a man needs to welcome and be welcomed, to love and be loved for himself, through genuine and deep interpersonal relationships that respect his autonomy". This subsequently generates the appearance of feelings as well as emotions, which allow the tourist to enter a different atmosphere, in which he will feel listened to, at ease and which could change his behavior. So, we can say that hospitality gives rise to the feeling of belonging to the community that Gouirand [33] considers as an element of tourist hospitality and for Oliver et al. [34] as "a positive emotional state that occurs when expectations are exceeded to a surprising degree", which leads to enchantment. For Cova [35], hospitality manifests itself in the desire to live off a new experience and the search for connections that guests no longer find in industrial hotels. Stringer [36] has also shown that owners often emphasize the friendly atmosphere of their homes in order to impress guests and acquire loyal customers. These customers have a human need in order to be warmly recognized and to "feel like

they are welcome" [37]. The latter two insist that the experiences that tourists have in a destination must be imbued with a "spirit of hospitality". This current of analysis, based on the opening of the "home" abroad could be a good indicator of perceived hospitality [35].

Like hospitality, enchantment presents itself as an important source in the social environment of a destination or place. For Bonnefoy-Claudet et al. [38], enchantment refers to a "feeling of satisfaction, joy, of exceptional quality, which grasps the being and transports him". Enchantment means the extent to which a person feels good, joyful, or happy towards a situation whose arousal refers to the extent to which an individual feels energized and active [39].

These two concepts of hospitality and enchantment remake the social environment that a client seeks during his stay. In a vector of self-development, interaction with other travelers or with members of the host community appears to be a motivation that leads to an extreme feeling of pleasure, which can lead to impact the process of the tourist decision-making. This contact presents a characteristic inherent to tourism, which allows the tourist to build a strong social link [40], to have a possibility of learning, and an exploration of places and cultures. Countless travelers seek the unknown of each place, which results in another sense of magical pleasure. Dayour [41] asserts that traveling to a place is an opportunity to meet and communicate with others and experience a magical emotion. In addition, based on personalized service, long-lasting relationships between companies and their customers will exceed expectations, leading to customer delight [42]. This confirms Hallberg's [43] research that social contact translates into the need to consume time with family and/or friends as well as the need to meet new people beyond the normal circle of acquaintances. While consumer delight has resulted in a better tool for increasing customer retention ratio.

*2.4. Involvement in the Experience*

There is a lack of consensus in the literature on the definition of the term involvement [44,45]. Celsi and Olson [46] perceived personal relevance as the essential characteristic of involvement, that is, involvement in an object, activity, or situation perceived as relevant [47]. Involvement is defined as "a state of unobservable motivation, excitement or interest" in a product or activity evoked by a stimulus or situation [46], and this affects perceptions of the value of the tourism experience [48]. The concept is associated with a lasting type of involvement, while the involvement situation is seen to reflect temporary feelings of involvement in a particular situation, which tend to have an impact before the purchase [49], and a moderating effect on the outcome of the tourism experience [50].

The concept of experience-involvement is presented as real-time personal involvement during the consumption of a given experience [47]. Some experiences can be very involved and involve emotions. Involvement is introduced as a tool for measuring experiences at a site [47]. Some researchers [51,52] have emphasized the role of involvement in the experience, but they have not specifically defined it as an involvement introduced into the experience. Others have studied similar phenomena without using the term implication of experience or differentiating it from the concept of flow [47]. Havitz and Mannell [53] measured the relationship between enduring and situational involvement, and flow. However, the flow experience is only a certain (the highest) dimension of the experience involvement. One of the main distinctions between involvement and experience involvement is that even different levels and types of involvement (e.g., enduring and situational involvement) influence consumer attention [53], the involvement of experience is the consequence [47]. The involvement in the experience does not define the purchasing decision, but the consumption of an offered service or experience plays an essential role in the development of the experience and the co-creation of value with the provider of services.

*2.5. Theory of Reasoned Action and the Theory of Planned Behavior*

The theory of reasoned action [54], just like the theory of planned behavior [54], aims to predict the behavior of individuals or to understand how a behavior can evolve

or change. Indeed, for the authors of these models, it is intention that constitutes the determining factor of behavior [55,56]. The stronger the intention, the more effort the person will put into this behavior and the more likely they are to engage in this behavior. Behavioral intention is determined, in whole or in part according to these theories, by the attitude of the individual towards that behavior. However, the theory of planned behavior provides an additional variable compared to the theory of reasoned action. This additional variable, which is perceived behavioral control refers to the perceived ease or difficulty in achieving the behavior. This behavioral control can be influenced by past experiences but also by anticipated obstacles. This variable can influence the onset of behavior in direct or indirect ways [55]. Thus, even if an individual has a favorable attitude towards a behavior and his entourage approves this behavior, he will not develop an intention to act if he does not believe that he has the necessary resources (time, money, skills, etc.) or does not believe that he has mastered the situation to get there. Adding the latter variable helps to explain why there may be a mismatch between behavioral intention and actual behavior. Reasoned Action Theory and Planned Behavior Theory have both considered an individual's intention to be a central factor in adopting a given behavior [57]. Thus, consumption intention can be considered as the predictor of behavior and of performing future behavior [55].

## 3. Research Model and Hypothesis Development

### 3.1. Experiential Consumption, Involvement in the Experience and Intention to Revisit

Experiential tourism motivates the tourists to always seek new destinations as well as more and more new experiences [58]. This is a relatively new approach, emphasizing the importance of searching for new impulses [59]. This new paradigm considers tourism as an opportunity for personal development and satisfaction of inherent human curiosity. In this case, the hallow effect works too [59]. A positive experience with guesthouses means a new motivation for searching for similar touristic places and services. Self-catering and traditional accommodations can be important drivers of tourism, especially for consumers, who would like to change their lifestyle in time of vacation [60]. Based on these considerations, we have formulated the hypotheses. In the following, the hypotheses are not ordered from 1 to 7 and this is due to our research, so they do not follow the chronological order:

**Hypothesis 1 (H1).** *Experiential consumption positively influences the involvement in an experience of staying in a guesthouse.*

**Hypothesis 2 (H2).** *Social environment positively influences the involvement in an experience of staying in a guesthouse.*

**Hypothesis 3 (H3).** *Involvement in the experience positively influences the intention to revisit a guesthouse.*

### 3.2. Experiential Consumption and Intention to Revisit

The concept of revisit intention refers to a customer's intention to continue using a product/service or return to a location and is an indicator of satisfaction [61]. It signifies the likelihood that a customer may return to the same location to consume the same experience in the future [61]. There are numerous results of research on the relationship between experiential consumption value and revisit intention [61,62]. The intention to revisit is a proxy of consumer satisfaction with the touristic product [63]. Consumers tend to get repeatedly involved in their accommodation experiences because they want to relive their former pleasant feelings [49,50]. Based on this consideration, we have developed the hypotheses as follows:

**Hypothesis 4 (H4).** *Experiential consumption positively influences the intention to revisit a guesthouse.*

**Hypothesis 5 (H5).** *Social environment positively influences the intention to revisit a guesthouse.*

*3.3. Social Environment and Involvement in the Experience*

It is well documented that the social environment significantly influences consumer behavior in experience-based tourism [47,50]. Certain social factors, such as enchantment and hospitality can lead customers to be highly involved and engaged in the experience. The concept of experiential value consumption [64] emphasizes the importance of the social environment of an experience as a facilitator of involvement of consumers into the experience. The staged experience creation [64] theory focuses on the formation and design of the social environment of an experience. This paradigm emphasizes that the social environment of the experience must have a theme and should be filled with cues, positively influencing the experience. All of the senses must be integrated into this experience because this will enhance the intensity of memory. Therefore, the following hypothesis is proposed:

**Hypothesis 6 (H6).** *Involvement in the experience mediates the relationship between experiential consumption and intention to revisit a guesthouse.*

**Hypothesis 7 (H7).** *Involvement in the experience mediates the relationship between social environment and intention to revisit a guesthouse.*

The whole hypotheses of this research work are demonstrated in detail in the theoretical model in Figure 1.

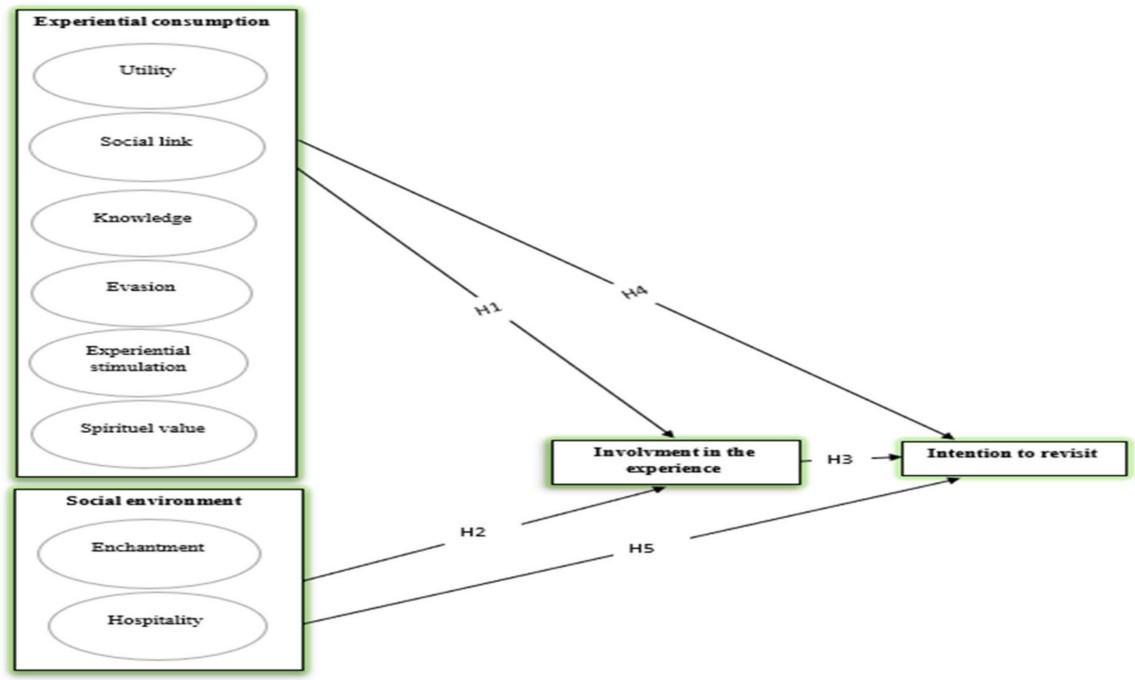

**Figure 1.** Theoretical framework (Source: Own edition, 2020).

**4. Materials and Methods**

*4.1. Study Context*

The crisis from which the tourist sector in Tunisia suffers is structural and dates back several years and the crisis of Covid-19 has aggravated the situation of tourism sector. Indeed, the tourist activity in Tunisia is limited to seaside tourism, which, for a long time,

has shown its limits. This situation has generated multiple adverse effects, including seasonal and precarious employment [65].

Thus, for Tunisia to enhance its tourism sector, it is imperative to turn to other products such as alternative tourism. To do this, some promoters of the sector have gone to the discovery of deep Tunisia and they have created projects based on alternative tourism. This tourism preserves the heritage and wealth that constitute the identity of Tunisia.

These new projects do not replace classic seaside tourism, but the idea was only to complement, enrich, and enhance it, through the development of the cultural heritage of the country, to promote the diversification of the tourism product, and to develop regional tourism.

Based on the evolution of the needs and tastes of consumers, the appearance of guest houses seems to take more and more place in the world tourism, and in Tunisia, in our context. By discovering gastronomy, nature, and the Tunisian people, the tourists will have experiences lived far from the limited spaces of the hotels. The establishment of guest houses in the Medina of Tunis and in the old cities throughout the Tunisian territory contributes to the revival of the craft industry and participates in the promotion of local products and gives another image different from daily life [65].

Despite the growth of the concept of guest houses and the dynamic role they play in promoting economic and social activity in Tunisia, especially in the face of the tourism and security crisis that shakes Tunisia, they rely on the reassuring side to attract travelers, but no clear strategy has been discussed or developed. In the absence of a state policy, investment incentives, and laws relating to this niche, promoters are faced with a double difficulty: obtain approval and find a source of funding (bank or other).

### 4.2. Sample and Data Collection

To test the conceptual model proposed in this work, an electronic survey was adopted. The population is represented by Tunisian tourists who have had at least one experience in a guesthouse in Tunisia.

Due to the COVID-19 pandemic, we have contracted with potential respondents via social media. The online data collection has been realized by Google Drive. Participants had had to answer all questions items before they could submit their survey, thereby eliminating the problem of missing values. At the end of the survey period (during the months of October, November, and December 2020), we had 259 filled out questionnaires.

The demographic characteristics of respondents are summarized in Table 1, the sampling error of this study was approximately 71.28% (95% confidence interval). Most of the variables had mean values greater than 5 (slightly agree). We have used 1–5 scales, indicating that respondents generally had very high scores for the constructs. The absolute skewness and kurtosis values for all variables were less than 1.5.

### 4.3. Research Instrument

Measurement scales were developed in the following process. First, a list of measurement items was identified based on a broad literature review. The elements used to measure the sixth dimensions of the value of experiential consumption were as follows: utility (four items), social connection (three items), knowledge (three items), escape (four items), experiential stimulation (four items), spiritual value (three items) (see Appendix A, Table A1), two dimensions of the value of the social environment (enchantment (five items) and hospitality (four items)) (see Appendix A, Table A2) each adapted from Aurier and al. [23] and Belhsen and Sentel [66], respectively. The involvement in the experience was measured by ten items from Zaichkowsky [67] (see Appendix A, Table A3). Lastly, intention to revisit was measured by using six items from Tosun and al. [68] (see Appendix A, Table A4). All the constructs were measured on a five-point Likert scale ranging from (1) strongly disagree to (5) strongly disagree. The questionnaire was translated into French and Arabic and then retranslated into English to prove the exactness of the connotation of each component [69] for local tourists.

**Table 1.** Demographic profile of the local tourists' respondents.

| | | | |
|---|---|---|---|
| **Gender** | Male | 150 | 57.9 |
| | Female | 109 | 32.1 |
| **Age** | 18–25 | 15 | 5.8 |
| | 26–34 | 33 | 12.7 |
| | 35–49 | 92 | 35.5 |
| | 50–64 | 82 | 31.7 |
| | Over 65 | 37 | 14.3 |
| **Monthly household income** | Lower than 1200 DT * | 25 | 9.7 |
| | 1200–2000 DT | 45 | 17.4 |
| | 2000–4000 DT | 87 | 33.6 |
| | More than 4000 DT | 102 | 39.4 |
| **Education** | High school or lower | 3 | 1.2 |
| | Post-secondary qualification | 44 | 17 |
| | Bachelor | 112 | 43.2 |
| | Master of higher | 100 | 38.6 |

* 1DT (Tunisian Dinar) = 0.31 EURO.

### 4.4. Analytical Methods

The demographic characteristics of respondents have been analyzed by frequency tables. A reliability test was performed using Cronbach's alpha and confirmatory factor analysis (CFA) was performed to assess construct validity. Reduction of variables had been carried out by CFA. Structural equation modelling (SEM) was used to identify structural relationships between the constructs, applied by the model. For both CFA and SEM, a two-step approach was applied to test the significance of each path coefficient [70]. The data were analyzed by application of SPSS 25.0 and SmartPLS 3.2 software.

## 5. Results and Findings

### 5.1. Demographics of Respondents

Analyzing the demographic characteristics of the respondents (See Table 1) it is obvious, that the sample is dominated by the middle or middle-upper class, middle-aged man, with a relatively high level of qualification. This part of the Tunisian population has a relatively high free disposable income. This fact can be considered as a favorable and informative one because this is the real target group of guesthouses: consumers who seek to enjoy life and embark on new experiences.

### 5.2. Factor Analysis

After consultation with the specialist group, a pilot study was carried out to improve the measurement procedure further. We pre-tested the questionnaire by questioning five local specialists, requesting them to evaluate the questionnaire and item statements; following this, we organized a larger pilot test with 30 participants to reformulate the items with the aim of reducing the sampling errors. A total of 259 completed questionnaires were collected from guests staying in guesthouses.

An exploratory factor analysis (EFA) was implemented using the pilot data. The EFA for constructs, including experience consumption, social environment, experience involvement, and intention to revisit, all reported KMO values higher than 0.8, and significance of Bartlett's test, indicating their suitability for factor analysis. However, in this initial EFA, four elements (Social link3, Knowledge3, Enchtment4, and Enchtment5) failed to load significantly on any dimension. Therefore, these elements were removed from further analysis. This, the reduced model yielded a four-factor structure (Table 2). This factor structure is in line with the theoretical propositions. Factor No. 1 includes elements Uty1 to Uty4, SL1 to SL2, KN1 to KN3, EV1 to EV4, ES1 to ES4 and SV1 to SV3, referring to experiential consumption (ES), and Factor No. 2 includes elements Encht1 to Encht3 and Hyp1 to Hyp4, which represent the social environment (SE). Factor No. 3 indicates the items Invt1 to Invt10 referring to the involvement in the experiment (Invt). Factor

No. 4 includes elements IR1 to IR6, denoting intention to revisit (IR) (see Appendix A, Tables A1–A4 for the abbreviations of the items).

**Table 2.** Reliability of measurement scales.

| Variables | Dimensions | Items | Factor Loading | Eigenvalues | % of Variance | Cronbach's Alpha |
|---|---|---|---|---|---|---|
| | Experiential consumption (KMO = 0.680; Test of Bartlett = Significant $p < 0.001$) | | | 1.045 | 79.241 | 0.917 |
| | Utility | Uty1 | 0.527 | | | |
| | | Uty2 | 0.575 | | | |
| | | Uty3 | 0.777 | | | |
| | | Uty4 | 0.802 | | | |
| | Social link | SL1 | 0.842 | | | |
| | | SL2 | 0.842 | | | |
| | Knowledge | KN1 | 0.714 | | | |
| | | KN2 | 0.718 | | | |
| | Evasion | EV1 | 0.594 | | | |
| | | EV2 | 0.788 | | | |
| | | EV3 | 0.860 | | | |
| | | EV4 | 0.811 | | | |
| | Experiential stimulation | ES1 | 0.907 | | | |
| | | ES2 | 0.851 | | | |
| | | ES3 | 0.831 | | | |
| | | ES4 | 0.858 | | | |
| | Spirit value | SV1 | 0.576 | | | |
| | | SV2 | 0.702 | | | |
| | | SV3 | 0.674 | | | |
| | Social environment (KMO = 0.759; Test of Bartlett = Significant $p < 0.001$) | | | 5.455 | 60.611 | 0.924 |
| | Enchantment | Encht1 | 0.793 | | | |
| | | Encht2 | 0.885 | | | |
| | | Encht3 | 0.821 | | | |
| | Hospitality | Hyp1 | 0.700 | | | |
| | | Hyp2 | 0.824 | | | |
| | | Hyp3 | 0.743 | | | |
| | | Hyp4 | 0.752 | | | |
| | Involvement in the experience (KMO = 0.724; Test of Bartlett = Significant $p < 0.001$) | | | 1.035 | 79.274 | 0.851 |
| | | Invt1 | 0.700 | | | |
| | | Invt2 | 0.585 | | | |
| | | Invt3 | 0.620 | | | |
| | | Invt4 | 0.639 | | | |
| | | Invt5 | 0.830 | | | |
| | | Invt6 | 0.750 | | | |
| | | Invt7 | 0.861 | | | |
| | | Invt8 | 0.925 | | | |
| | | Invt9 | 0.837 | | | |
| | | Invt10 | 0.612 | | | |
| | Intention to revisit (KMO = 0.646; Test of Bartlett = Significant $p < 0.001$) | | | 3.289 | 54.811 | 0.867 |
| | | IR1 | 0.585 | | | |
| | | IR2 | 0.657 | | | |
| | | IR3 | 0.536 | | | |
| | | IR4 | 0.581 | | | |
| | | IR5 | 0.595 | | | |
| | | IR6 | 0.685 | | | |

Following the phase of purification and verification of the psychometric qualities of the measurement scales of the constructs of the model, we chose the method of structural equations as the method of data analysis for the interpretation of the results of our study,

to perfect the analysis of the data collected, estimate the models, and test the research hypotheses. The structural equation modeling approach that best suits the nature of our data, and that we have taken, is that of Partial Least Squares (PLS).

In our model, there were four latent variables (constructs): (1) social environment, (2) experiential consumption, (3) involvement inexperience and (4) intention to revisit. The latent variables (1) and (2) are second-order exogenous latent variables. The latent variable (3) can be considered as a mediating variable because this is a link between different latent variables in the model. These four latent variables have been considered as reflective first-order constructs. We have aimed to estimate these variables according to the confirmation criteria of the reflexive constructs.

### 5.3. Assessment of Model Using Partial Least Squares Structural Equation Modeling

5.3.1. Assessment of Measurement Model

Confirmatory Factor Analysis (CFA) was initially performed using SmartPLS software with 259 response data. To assess internal consistency, we check the composite reliability, the reliability of the indicators (reliability of the items) as well as the Cronbach Alpha following the recommendations of Hair and al. [71].

As a rule of thumb, the composite reliability coefficient represented by the construct respecting the correlation of its items must generally be greater than (0.7) and less than (0.95), with the tolerance of a minimum threshold of (0.6) [71]. Similarly, for Cronbach's alpha, which is interested in the correlation of items between them and their representativeness of the same construct, its value must be greater than (0.7). As for the reliability of the indicators, it concerns the elements of the constructions, and it is verifiable through the external loads of the indicators (external loading). According to the recommendations of Hair et al. [71], elements with factor loading values lower than 0.4 should be removed, the elimination of items whose loadings are between (0.4) and (0.7) must be done by checking its impact on the improvement of indicators (CR) and the average of the extracted variance (AVE). By examining the loading of indicators, we identified some items with low loading values. Following the recommendations of Hair et al. [71], we proceeded by eliminating them one by one, starting with the element that represents the value of the lowest load factor and checking its impact on the reliability of the balance. The elimination of constructs of "utility" (Utly_1 = 0.276, Utly_2 = −0.124 and Utly_3 = 0.456), "knowledge" (KN_8 = 0.542), "experiential stimulation" (ES_15 = 0.557 and ES_16 = 0.482), and "Involvement in the experience "(Invt_8 = -0.124 and Invt_10 = 0.074) improved the values of the reliability indicators of the CR and AVE scales. After the simplification of the model, the verification of the loading and the CR and AVE indices showed good reliability of the indicators, because all loadings were greater than 0.5 and the measurement scales all have good internal consistency. CR was higher than 0.6 and less than 1 (See Table 3).

**Table 3.** PLS-SEM assessment results of reflective measurement models.

| variables | Cronbach's Alpha | Rho_A | Composite Reliability | Average Variance Extracted (AVE) |
|---|---|---|---|---|
| Experiential consumption | 0.878 | 0.930 | 0.915 | 0.730 |
| Intention to revisit | 0.870 | 0.884 | 0.901 | 0.604 |
| Involvement in the experience | 0.900 | 0.905 | 0.921 | 0.626 |
| Social environment | 0.858 | 0.860 | 0.934 | 0.876 |

For further evaluation, the extracted mean variance (AVE) was examined. The AVE values of the latent variables were greater than the recommended value of 0.50 indicating that the construct explains more than 50% of the variance of its components (directly measured variables) [71].

The discriminant validity of the model has been tested by Fornell and Larcker [72] method (Table 4). The criterion of acceptance of the discriminant validity according to this

approach is the satisfaction of the condition, that the square root of each latent variables' AVE should be higher than its highest correlation with any other model (construct).

**Table 4.** Square root values of AVEs and Fornell–Larcker discriminant validity test.

| Variables | Experiential Consumption | Intention to Revisit | Involvement in the Experience | Social Environment |
|---|---|---|---|---|
| Experiential consumption | 0.854 | - | - | - |
| Intention to revisit | 0.748 | 0.777 | - | - |
| Involvement in the experience | 0.553 | 0.748 | 0.791 | - |
| Social environment | 0.867 | 0.726 | 0.622 | 0.936 |

Thus, the convergence validity has been obtained. Once reliability and convergent validity were demonstrated, it was decided to use the most recommended and popular criterion to evaluate the discriminant validity of the combinations studied that should be specified in the model as well as an empirical difference with other formulations rather than the less rigorous cross-loading approach [63]. Discriminant validity must be formed to prove the difference between concepts. In this perspective, several criteria can be requested for the assessment of the discriminant validity [73]. Corresponding to the current literature, the two most fundamentalist methods to evaluate the discriminant validity are the heterotrait-monotrait ratio (HTMT) and the Fornell–Larcker criterion [72]. Consequently, we employed both methods to evaluate the discriminant validity in this research paper. The HTMT value for the whole concepts must be less than 0.9 to confirm the discriminant validity founded on the HTMT approach [74].

Furthermore, to determine the discriminant validity founded on the Fornell–Larcker criterion, the square root of the AVE of each construct must be greater than its correlation with the other constructs of the model [72]. The results presented in Table 4 demonstrate an adequate discriminant validity founded on both approaches. In the second step, we recognized the values of experiential consumption and the social environment as second-order formative constructs applying the score of their correlated dimensions from the first step [75]. Utility, social connection, knowledge, escape, experiential stimulation, and spiritual value established the value of experiential consumption [23,76], while enchantment and hospitality established the concept of the second-order social environment. Consequently, in the second step, the context of this research contains reflective constructs.

### 5.3.2. Assessment of the Structural Model

The structural model has been assessed based on a complex set of criteria. The structural model was assessed based on the guidelines recommended by Hair et al. [72]. The standard evaluation criteria considered are the evaluation of collinearity, statistical significance and relevance of the path coefficients, the cross-validated redundancy measure based on the blindfold $Q^2$, the coefficient of determination ($R^2$), and the predictive model used by the PLSpredict procedure.

It is well documented that the collinearity between different variables lead to a singular matrix, and this causes considerable problems in further steps of evaluation of the model. That is why the assessment of collinearity is a critical, first step of model evaluation.

The collinearity has been measured by the variance inflation factor (VIF) (See Table 5). This is the reciprocal value of the share of variance, which is not explained by indicators in a given block. In general practice, the VIF should be lower than 5 to continue the calculations, without removing one or more indicator variables. In our case, the VIF was lower than this limit. This fact has been proven, that the indicator variables were not redundant, and each of them could be kept for further calculations because they have furnished a piece of information, which could have not been obtained by application of a smaller set of indicators.

**Table 5.** Evaluation of the Variance Inflation Factor (VIF).

| Variables | Experiential Consumption | Intention to Revisit | Involvement in the Experience | Social Environment |
|---|---|---|---|---|
| Experiential consumption | - | 4.046 | 4.041 | - |
| Intention to revisit | - | - | - | - |
| Involvement in the experience | - | 1.632 | - | - |
| Social environment | - | 4.579 | 4.041 | - |

A critical part of the model analysis is the calculation and evaluation of path coefficients because the majority of hypotheses will be proven or rejected as a result of these investigations.

The generally used criteria for acceptance of a path coefficient in two-tailed tests is 1.65 at 10%, and 1.96 at a 5% two-tail significance level. Analyzing the results (Table 6), it is obvious that their paths experiential consumption →intention to revisit, involvement in the experience →intention to revisit, and social environment →involvement in the experience were significant, but in two cases, experiential consumption →involvement in experience and social environment →intention to revisit could not have been proven significant relationship.

**Table 6.** Final path coefficients.

| Paths | Original Sample (O) | t-Statistics (∣O/STDEV∣) | *p*-Values | 5.0% | 95.0% |
|---|---|---|---|---|---|
| Experiential consumption →intention to revisit | 0.451 | 6.274 | 0.000 | 0.334 | 0.569 |
| Experiential consumption →involvement in the experience | 0.055 | 0.485 | 0.627 | −0.127 | 0.244 |
| Involvement in the experience →intention to revisit | 0.473 | 8.580 | 0.000 | 0.373 | 0.557 |
| Social environment →intention to revisit | 0.040 | 0.592 | 0.554 | −0.075 | 0.149 |
| Social environment -> involvement in the experience | 0.574 | 5.609 | 0.000 | 0.393 | 0.726 |

In the case of mathematical modelling of complex processes, it is always an important question whether the model fits the data "by chance", or the logical construction is capable to uncover some really important relationship between different directly or indirectly measured hidden variables. This question can be answered by the application of the model for forecasting. The predictive relevance of the model has been analyzed based on $Q^2$ value, calculated by blindfolding method (Table 7).

**Table 7.** Summary of the evaluations of the predictive relevance $Q^2$ and of the size effect $q^2$.

| Constructs | $Q^2$ Includes (Predictive Importance) | $Q^2$ Excluded | $q^2$ (Size Effect) | Quality of the Size Effect |
|---|---|---|---|---|
| Experiential consumption →intention to revisit | 0.406 | 0.379 | 0.045 | Weak |
| Experiential consumption →involvement in the experience | 0.227 | 0.231 | −0.005 | Weak |
| Involvement in the experience →intention to revisit | 0.406 | 0.330 | −0.149 | Medium |
| Social environment →intention to revisit | 0.406 | 0.410 | −0.006 | Weak |
| Social environment →involvement in the experience | 0.227 | 0.180 | 0.061 | Weak |

As a general rule, if the Q2 is larger than 0.02 the predictive relevance of the model on path coefficient can be supposed. On this basis, it can be concluded that all of the path coefficients can be forecasted based on the model, but the efficiency of the model on forecasting is different. According to [71], if the $q^2$ value is larger than 0.02 the exogenous

construct has weak, if this value is between 0.03 and 0.15 is a median, and if it is greater than 0.35 it has a large predictive effect. Following this guideline, we can conclude that the predictive relevance of involvement in the experience is medium-sized on intention to revisit variable. The predictive relevance of the model from point of view between different constructs has been relatively weak, but the effect of involvement in the experience on intention to revisit has been medium.

The model fitting can be further analyzed based on the $f^2$ value (Table 8). This indicator quantifies the effect of an endogenous construct. The limits of weak, medium, and large size effects are the same as in the case of $q^2$ value. In this case, we see a large effect on involvement in the experience and intention to revisit. The effect of experiential consumption on intention to revisit is moderate, in all other cases it is small. The effect of social environment on involvement in the experience is near to the classification of this effect of models' predictive accuracy and relevance as moderate.

**Table 8.** Effect sizes of the exogenous constructs on the model's predictive accuracy and relevance.

| Constructs | Original Sample (O) | Sample Mean (M) | Standard Deviation (STDEV) | t-Statistics (|O/STDEV|) | *p*-Values | Effect Size |
|---|---|---|---|---|---|---|
| Experiential consumption →intention to revisit | 0.180 | 0.186 | 0.065 | 2.790 | 0.005 | Moderate |
| Experiential consumption →involvement in the experience | 0.001 | 0.007 | 0.009 | 0.133 | 0.894 | Small |
| Involvement in the experience →intention to revisit | 0.492 | 0.506 | 0.136 | 3.605 | 0.000 | Large |
| Social environment -> intention to revisit | 0.001 | 0.005 | 0.007 | 0.192 | 0.848 | Small |
| Social environment -> involvement in the experience | 0.133 | 0.140 | 0.053 | 2.502 | 0.012 | Small |

Based on these elements, Henseler et al. [77] call for using "RMStheta", which emerged in 1989 with Lohmôller without being applied in the context of the PLS-SEM, as an alternative measure of model fit. Its application should be done with caution in relation to low threshold values because a weak fit does not necessarily imply a poor predictive power of the model. Given that the threshold values for this indicator have not yet been specified and that these measures lack in-depth research in the PLS-SEM context, Henseler et al. [77] invite accepting higher values by stipulating that the fixed threshold is too low for PLS-SEM. Starting from the specificities of the PLS-SEM, which focuses on prediction rather than on explanatory modeling, some researchers are calling for the use of evaluation criteria that reinforce its predictive nature. In this case, they advise against the use of current statistics, while waiting for other indicators to emerge. According to these researchers, these statistical tests are of little value and that using them can even affect the predictive power of the model. In our research, the value of the RMS theta (0.13) shows a result slightly higher than the threshold, which is a priori acceptable, and which leads us to conclude that the model has an acceptable predictive relevance (See Table 9).

**Table 9.** FIT model goodness-of-fit index (SRMR).

| SRMR Saturated | Estimated SRMR | RMS Theta |
|---|---|---|
| 0.120 | 0.120 | 0.130 |

As a summary, it can be concluded that our findings support the hypotheses on the importance of experimental consumption and social environment on involvement in the experience. These facts prove the H1 and H2 hypotheses. The results of this study support the direct effects of experiential consumption and social environment on intention to revisit (H4 and H5). The indirect effect of these factors on intention to revisit has been proven too,

via involvement in the experience (H6 and H7). The mediator effect has been determined by the bootstrap resampling method [78]. The result showed the significant mediating role for the involvement in the experience between the value of experiential consumption and the value of the social environment, and the intention to revisit. Hence, the findings of the present research emphasize the significance of the direct and indirect effects of the value of experiential consumption and the social environment on involvement in the experience and on the intention to revisit. Additionally, the findings revealed a significant effect of involvement in the experience on the intention to revisit guesthouses in Tunisia (H3).

### 5.4. Discussion and Implications

The current study studied the direct effects of the subdimensions of the customer's perceived experiential value—experiential consumption value, and social environment value—on involvement in the tourist experience (H1–H2) and on the intention to revisit (H4–H5); and the direct effect of involvement in the experience on tourists' intentions to revisit (H3). The results showed that the value of the social environment has a significant effect on the involvement in the tourist experience ($\beta = 0.574$, t = 5.609 > 1.96) but an insignificant effect on the intention to revisit ($\beta = 0.040$, t = 0.592), which appears to be opposed to the studies of Bonnefoy-Claudet L. et al. [39]. For the value of experiential consumption, it has a significant effect on the intention to revisit ($\beta = 0.451$, t = 6.274 > 1.96) that confirm the studies of Aurier et al. [23] and Jin et al. [19] but an insignificant relation with the involvement in the experience ($\beta = 0.055$, t = 0.485). Interestingly, each dimension of the value perceived by the tourist has a different effect. The results indicate that tourists who have stayed in traditional guesthouses enjoyed fun, relaxing, and original experiences that can facilitate the emergence of revisiting intentions while the enchantment and hospitality of staff contributes to their feeling of involvement in the experience. Same researchers have discovered a significant relationship between perceived experiential value [79] and intention to revisit [80]. The results confirmed the relationship between involvement in the experience of tourists and their intentions to revisit (H3) ($\beta = 0.473$, t = 8.580 >1.96). This would indicate that tourists who have stayed in guesthouses are fully involved in their guesthouse experiences and intend to return soon. For example, Brown G. et al. [80] explain that tourists who enjoy their stay in a guesthouse become more involved in their experience and, therefore, become more apt to revisit the place and proposed it to others. The results showed that the value of experiential consumption played an important role as predictors of intention to revisit the same guesthouse. In addition, tourists' connections with the staff in the guesthouse and the host community influence their emotions of enchantment. Therefore, the interaction values of the social environment have established positive emotions of experiences for tourists via the contact with other individuals, which contributes to the feeling of involvement in the experience that they are living. While preceding researchers have examined the relationship between tourist perceived value, social environment, and involvement in the experience, only some researchers have verified the interrelationships between the numerous subdimensions of experiential consumption that bring together the value of consumption and the social environment of the guests in the hospitality industry and the role of involvement in the experience, which appears in most of the research as a moderating variable. This study examined the interrelationship between involvement in the experience and intention to revisit (H3). The results supported the interrelationship between them. According to these results, involvement in the experience that a tourist has while staying in a guesthouse influences the feeling of coming back to the same place. This research paper examined the mediating role of involvement in experience in the relationships between consumption value and the social environment and the intention to revisit (H6 and H7). The results showed a significant mediating role for the value of involvement in the experience, indicating that enchantment and hospitality during guesthouse stay play an important role in intention to revisit ($\beta = 0.927$, t = 4.320 > 1.96). The results of the mediation evaluation showed that the increased sense of enchantment and hospitality of tourists in guesthouses increases involvement in the experience and facilitates the development of

intentions to revisit. The results of the mediation assessment showed that most of the effect of social environment value on revisit intention was transferred through involvement in the experience. However, the mediating role of experience involvement in the relationship between consumption value and intention to revisit (H7) is not significant (β = −0.122, t = 0.496). The findings of the mediation assessment revealed that the increase in the feeling of experiential consumption of the tourists in the guesthouses does not automatically increase the involvement in the experience but it does facilitate the development of the intentions to revisit.

This study has theoretical and practical contributions. For theoretical contributions, the construction of tourist perceived experiential value has been widely examined in the hospitality industry [23,25], the dimensions of tourist experiential consumption and social environment have not been well studied [23,25,65].

The findings of this research showed a considerable contribution to the literature, describing the important role of these dimensions of perceived experiential value on the tourist staying in a guesthouse. Second, previous studies have measured the experiential value perceived by the tourist as a multidimensional and one-dimensional concept and investigated its relationship with involvement in the experience and intention to revisit. This study examined the effect of experiential consumption and social environment as perceived by the tourist on the involvement in an experience and the intention to revisit. Experiential consumption and the social environment of guesthouses is a factor influencing guest involvement. Although the importance of experiential consumption and the social environment has already been recognized, there has been no systematic development of the application of this concept to guesthouses and this study contributes to deepening our understanding of the role of experiential consumption and the social environment in the guesthouse sector. In dealing with experiential consumption and the social environment as a whole, we investigated its impact on involvement in an experience and intention to revisit, although different factors of the concept may impose different effects. The results of our study offer new perspectives. Tertiary, only some researchers have assessed the role of experience involvement value as a mediating variable [47,67] in the relationship between experiential consumption value and social environment value on intention to revisit, particularly in the guesthouse framework. This research goes ahead to study the mediating role of involvement in experience in the relationships between the value of experiential consumption and the value of the social environment on the intention to revisit. The results confirm that involvement in the experience plays a mediating role in the relationship, which combines the value of the social environment on the intention to revisit. The results showed that the relationship between the value of experiential consumption and the intention to revisit has no effect. Therefore, involvement in the experience is an important element in developing the intention to revisit among tourists.

The current study developed a conceptual model of the factors that lead to customer return to guesthouses, taking into account experiential consumption and the social environment. The resulting framework can help academics and managers better understand the key factors impacting the post-purchase behaviors of customers who frequent non-standard accommodation types, whose consume an experiential consumption and social environment perspective. The results of this research could be applied to other emerging markets or developing countries.

According to the results of this study, the value of experiential consumption and the value of the social environment have significant effects on involvement in the tourist experience in Tunisian guesthouses. However, the value of the social environment (including enchantment and hospitality) has the greatest effect on involvement in the experience and, therefore, should be emphasized in the management of guesthouses. In addition, the results clarify the importance of the value of experiential consumption in the development of intention to return to guesthouses. Second, this study found that the experiential consumption and the social environment perceived by the tourist and involvement in the experience are essential components for tourists, which inform their intention to return to

the guesthouse. Preceding researchers have shown that tourists' decision to revisit a specific place is largely based on their satisfaction [75] regarding this place during their previous visits, but our research has shown that their decision also depends on the involvement that a client may experience during their experience in a guesthouse. Hence, it is necessary that managers aim to involve their tourists by fostering long-term collaborative relationships with them, as well as offering them new and distinctive experiences. Such measures should cause a boosted sale and enhance the guesthouse performance. The owners and managers of guesthouses should intend to establish a suitable marketing strategy, such as smart pricing strategies, to offer to their customers a superior value that put them ahead of their competition. This would permit guesthouses to offer tougher competition through price differentiation, improving guest retention, and providing them with unique benefits. While these are issues that concern most tourists, room prices play an indisputably significant role in the predicting accommodation demand. The clients are more expected to make choices related to their stay founded on the perceived experiential value of potential gains and losses, often gauging value utilizing shows such as price, brand, or electronic word of mouth. In future research, it is best to study these factors as a trigger for return intention for guesthouses.

Other parameters that guesthouses should take into consideration too involve the guesthouse reviews, which for several customers are dependent on promotional price offers, hygiene of the guesthouse, quality of hotel services, as well as bar and restaurant facilities, especially now with the pandemic situation. Due to the strong contests in the hospitality market, the expansion of the guesthouse demand depends on understanding how tourists choose to stay to spend time away from everyday life. Providing your clients with exceptional profits can also support guesthouses to retain their client relationships, thereby improving their intentions to return to visit. Those who choose to stay in a guesthouse tend to seek new experiences; however, the interests and experiences of these visitors may vary from each other. Remarkably, the findings of this research paper reveal that the visitor connections with the host community and hotel staff play an essential role in engaging tourists' retirement experience and that the secret to these interactions may be a unique feature of this traditional form of housing compared to hotels, resorts, and Airbnb. Consequently, we recommend that innkeepers, together with local communities, try to build an appealing tour packages aimed at raising tourists' intentions and strengthening cooperation between travel agencies. Local tourism authorities and restaurants can support the parts to make an unforgettable experience and encourage them to repeat visits. Pensions may also seek to provide value-added services, such as travel guides, to better meet the needs of local tourists. Most traditional guesthouses are in places of historical, cultural, or aesthetic interest, usually far from typical accommodation facilities. Therefore, providing better access to tourists with good infrastructure can improve employment rates and improve overall pension benefits. Another strategy that a pension could follow is to develop attractive and interactive websites, thus, giving pensions an increased remote presence to promote hotel packs, intermingle with prospective customers, and expedite the processes of decision-making.

## 6. Conclusions

The current global tourism context, caused by the outbreak of COVID-19, is profoundly disrupted. The mobility of tourists, no matter what scale, is stopped. Closed borders, interrupted air, sea, rail, and road transport, canceled tourist activities, and forbidden tourist accommodation are all obstacles to the reign of Homo turisticus. As early as March 2020 and for many weeks, the global tourism system has stalled, with the most optimistic observers not seeing a restart for several months. The worst-case scenarios, on the other hand, simply do not envisage a massive return of tourists before 2021, if the health context allows it. Therefore, it is important for each country to find other sources to revive the sector again. The experiential approach for local tourists is seen as a source of change for the local tourism of a country. Indeed, although the importance of the quality of service

has been highlighted in the tourism literature, another related and nuanced factor, that of the value of an experiential consumption towards a service. In addition, experience has an important influence on the evaluation of consumers and their involvement in a given service. As a result, managers then need to understand the experiential phenomenon in order to develop tourism products/services and reactivate the tourism sector again. This research investigated the role of experiential consumption and the social environment on the intention to revisit a guesthouse. This study leads on the one hand to enrich the knowledge of the role of experiential consumption and the social environment on the intention to return, and on the other hand, helps and responds to the problems that managers of guest houses suffer from and to the marginalization of this sector, especially now with the Covid-19 crisis.

### 7. Limitations and Future Research

This research paper has some limitations. The complexity of this question necessitates the due consideration of all facts in the generalization of these results. Further study should be carried out to uncover the relationship between accommodation and tourist destinations to better comprehend the influence of the tourist's perceived experiential consumption and social environment on tourists' participation in the experience and their revisit intention, as well as their interrelationships. Most of the respondents in this study were local tourists because of the health situation in 2020. However, a larger sample composed mainly of international tourists will provide important information and allow the results to be generalized. The forthcoming study should seek to assess the experiences of local and international tourists residing in guesthouses, as well as comparing their experiences with those of other kinds of accommodation. Furthermore, forthcoming research may take into consideration to explore the role of other variables, such as tourist satisfaction, commitment and loyalty, along with the intention of revisiting a guesthouse.

**Author Contributions:** Conceptualization, N.K.; methodology, N.K.; validation, Z.L.; investigation, N.K.; data analysis, N.K.; writing—original draft preparation, N.K.; writing—review and editing, Z.L.; visualization, Z.L.; supervision, Z.L. All authors have read and agreed to the published version of the manuscript.

**Funding:** This research received no external funding.

**Institutional Review Board Statement:** Not applicable.

**Informed Consent Statement:** Not applicable.

**Data Availability Statement:** Excluded.

**Acknowledgments:** This research is supported by the Hungarian University of Agriculture and Life Sciences (MATE), and Stipendium Hungaricum Scholarship.

**Conflicts of Interest:** The authors declare no conflict of interest.

### Appendix A

**Table A1.** List of items used to measure the experiential consumption.

| | | |
|---|---|---|
| | Uty 1 | I like it when everything is accessible and convenient in a guesthouse |
| | Uty 2 | I appreciate that the location of the guesthouse is well maintained and marked out |
| **Utility value (Uty)** | Uty 3 | The quality or location of the accommodation is important to me |
| | Uty 4 | I appreciate that the guesthouse offers many services (sports activities, local cuisine, ...) |

**Table A1.** *Cont.*

| | | |
|---|---|---|
| **Social link (SL)** | **SL1** | Staying in a guest house gives me the opportunity to talk about it with friends from another culture |
| | **SL2** | Staying in a guest house, I like it |
| | **SL3** | I like to make new contacts with people from another country and culture |
| **Knowledge (KN)** | **KN1** | Staying in a guesthouse is an opportunity to get to know new things |
| | **KN2** | Deep immersion in the culture of this country |
| | **KN3** | I got to know a new culture and tradition |
| **Evasion (EV)** | **EV1** | I like the feeling of freedom that the mountains give me |
| | **EV2** | During my stay in a guest house, I escape and forget the daily life |
| | **EV3** | I like to stay in a guest house because it is a real change of scenery |
| | **EV4** | I appreciate coming to a guest house because I always discover new landscapes, new experiences and new people |
| **Experiential stimulation (ES)** | **ES1** | I appreciate the stays in the guest house because they are the occasion to practice many activities |
| | **ES2** | I like being in a guest house because it is synonymous with various leisure activities |
| | **ES3** | I enjoy having many different experiences while staying in a guest house |
| | **ES4** | I like to learn new activities during my stay (biking, riding, ...) |
| **Spirit value (SV)** | **SV1** | I feel like I am expressing who I am when I stay in a guest house |
| | **SV2** | Staying in a guest house allows me to express who I am |
| | **SV3** | My personality counts a lot in the choice of my holiday accommodation |

**Table A2.** List of items used to measure the social environment.

| | | |
|---|---|---|
| **Enchantment (Encht)** | **Encht1** | I felt enchanted at one point during my visit to this guesthouse |
| | **Encht2** | I felt joyful at one point during my visit to this guesthouse |
| | **Encht3** | I felt exhilarated at some point during my visit to this guesthouse |
| | **Encht4** | Overall, I can say that I was amazed by this stay |
| | **Encht5** | Overall, I can say that I was transported by this stay |
| **Hospitality (Hyp)** | **Hyp1** | Very courteous, welcoming and friendly staff |
| | **Hyp2** | Extraordinary and warm welcome |
| | **Hyp3** | Dedicated staff is endearing, so much human warmth and Tunisian hospitality are present |
| | **Hyp4** | Values of sharing culture and traditions |

**Table A3.** List of items used to measure the involvement in the experience.

| | | |
|---|---|---|
| **Involvement in the experience (Invt)** | **Invt1** | A stay in a guest house is something that is important |
| | **Invt2** | A stay in a guest house is something that is interesting |
| | **Invt3** | A stay in a guest house is something that concerns me |
| | **Invt4** | A stay in a guesthouse is something that is exciting |
| | **Invt5** | A stay in a guesthouse is something that has great meaning |
| | **Invt6** | A stay in a guesthouse is something that is attractive |
| | **Invt7** | A stay in a guesthouse is something that is fascinating |

**Table A3.** *Cont.*

| | |
|---|---|
| **Invt8** | A stay in a guesthouse is something that is valuable |
| **Invt9** | A stay in a guesthouse is something that is involving |
| **Invt10** | A stay in a guest house is something I need |

**Table A4.** List of items used to measure the intention to revisit.

| | | |
|---|---|---|
| **Intention to revisit (IR)** | **IR1** | We will return once again to this guest house |
| | **IR2** | We will not fail to return as soon as the opportunity arises |
| | **IR3** | We want to stay here for a long time |
| | **IR4** | For sure, we will come back to spend the next vacations |
| | **IR5** | To come back without hesitation |
| | **IR6** | I want to come back as soon as possible |

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
