# Peer review of "Influence of Experiential Consumption and Social Environment of Local Tourists on the Intention to Revisit Tunisian Guesthouses: Mediating Role of Involvement in the Experience"

_sustainability, doi:10.3390/su13126584_

Round 1

Reviewer 1 Report

Title. The empirical research of the paper worked with national tourists in a certain type of accommodation, specific to Tunisia, so it is necessary that the title reflects that the tourists are nationals; since the results and conclusions cannot be extrapolated to all types of tourists.

Empirical study. It is necessary to reflect a table with technical data of the study. Containing the type of random sampling that was carried out, the level of confidence, the dispersion, sampling error and the period in which the data were collected.

In addition, the questionnaire must be specified.

Author Response

Thank you to the reviewers who consulted my work and added remarks that are all beneficial to approve the article.

We have taken into consideration all the remarks of the three reviewers, and we have made the necessary changes.

For reviewer 1, we have modified the title and the table with the technical data of the study. 

Reviewer 2 Report

322-323 This study uses a quantitative research approach with a survey questionnaire to collect data from 322 respondents - clients staying for a period in guesthouses scattered throughout Tunisia. – Regarding collecting data, the area you mentioned is still ambiguous. Are you talking about whole areas of Tunisia? Or some places in Tunisia. I recommend you elaborate on specific collected area. In addition, it might be better to provide the time of collection date or period.

343-346 The elements used to measure the sixth dimensions of the value of experiential consumption (utility [four items], social connection [three items], knowledge [three items], escape [four items], experiential stimulation [four items], spiritual value [three items],).

360-361 However, in this initial EFA, four elements (LS3, KN4, Encht4, and Encht5) failed to load significantly on any dimension. Therefore, these elements were removed from further analysis.

Although four elements were failed to use in this model, I recommend you provide these questionnaire sentences in the paper. In addition, under items, you just used abbreviation only. I recommend you provide whole or partial sentences of questionnaire items.

  1. Table 2: Demographic profile of the respondents

- Do we have any chances to identify the guests’ ethnicity? Are those all Tunisians?

Author Response

Thank you to the reviewers who consulted my work and added remarks that are all beneficial to approve the article.

We have taken into consideration all the remarks of the three reviewers, and we have made the necessary changes.

For reviewer 2, we modified the collected data to make it more visible and clearer. We added a table that explains the variables with their items and the abbreviations of each. Regarding the demographic profile remark, we have re-explained it to make it clearer for future readers.

Reviewer 3 Report

  1. The “research hypotheses” should be many literatures to support authors opinion. The research architecture of manuscript, please provide and explanation it. It is recommended to explain the research hypotheses one by one. The “Hypotheses Development” should be many literatures to support authors opinion.
  2. How did this research framework of manuscript develop process, please provide and explanation.
  3. This figure 1 is very mistiness and incorrect. Especially H6 and H7 are marked incorrectly.
  4. Questionnaire design is the important of quantitative research. Sampling methodology and techniques adopted are not clear. In consumer research, sampling details are crucial. The representativeness of the questionnaire has influence on the research results.

5.The citations references are very old and fall short of the journal’s expectation (Only 10 articles after 2019).

Author Response

Thank you to the reviewers who consulted my work and added remarks that are all beneficial to approve the article.

We have taken into consideration all the remarks of the three reviewers, and we have made the necessary changes.

For reviewer 3, we modified the figure of our conceptual model. For sampling methodology and techniques adopted, we have taken into consideration the remarks and reworked them to be more visible.
